# The Osteocyte with SB216763-Activated Canonical Wnt Signaling Constructs a Multifunctional 4D Intelligent Osteogenic Module

**DOI:** 10.3390/biom14030354

**Published:** 2024-03-15

**Authors:** Jinling Zhang, Ying Zhang, Jiafeng Chen, Weimin Gong, Xiaolin Tu

**Affiliations:** Laboratory of Skeletal Development and Regeneration, Institute of Life Sciences, Chongqing Medical University, Chongqing 400016, China; 2021110818@stu.cqmu.edu.cn (J.Z.); 2021130118@stu.cqmu.edu.cn (Y.Z.); 2023130462@stu.cqmu.edu.cn (J.C.); 2020111811@stu.cqmu.edu.cn (W.G.)

**Keywords:** osteocyte, SB216763, DOME, Wnt signaling, osteogenesis, 4D intelligent osteogenic module

## Abstract

The enhancement of bioactivity in materials has become an important focus within the field of bone tissue engineering. Four-dimensional intelligent osteogenic module, an innovative fusion of 3D printing with the time axis, shows immense potential in augmenting the bioactivity of these materials, thereby facilitating autologous bone regeneration efficiently. This study focuses on novel bone repair materials, particularly bioactive scaffolds with a developmental osteogenic microenvironment prepared through 3D bioprinting technology. This research mainly creates a developmental osteogenic microenvironment named “DOME”. This is primed by the application of a small amount of the small molecule drug SB216763, which activates canonical Wnt signaling in osteocytes, promoting osteogenesis and mineralization nodule formation in bone marrow stromal cells and inhibiting the formation of adipocytes. Moreover, DOME enhances endothelial cell migration and angiogenesis, which is integral to bone repair. More importantly, the DOME-PCI3D system, a 4D intelligent osteogenic module constructed through 3D bioprinting, stably supports cell growth (91.2% survival rate after 7 days) and significantly increases the expression of osteogenic transcription factors in bone marrow stromal cells and induces osteogenic differentiation and mineralization for 28 days. This study presents a novel approach for bone repair, employing 3D bioprinting to create a multifunctional 4D intelligent osteogenic module. This innovative method not only resolves challenges related to shape-matching and biological activity but also demonstrates the vast potential for applications in bone repair.

## 1. Introduction

Recent studies have shown that combining 3D printing with biomolecular materials notably improves the shape-matching and bioactivity of bone repair materials. Composite materials containing tissue cells, growth factors, and other materials can enhance the biological activity and biocompatibility of the materials themselves [1]. For example, materials loaded with bone morphogenetic proteins (BMPs) can stimulate osteoblast differentiation of mesenchymal stem cells [2,3], while vascular endothelial growth factor (VEGF) promotes the branching of existing blood vessels in the surrounding native tissue [4]. However, the rapid release of biofactors and improper positioning can lead to side effects such as bone resorption, swelling, and heterotopic ossification [5]. Therefore, the development of safe bioactive materials continues to be crucial.

In 2013, Skylar Tibbits of the Massachusetts Institute of Technology first proposed the concept of 4D printing [6], “4D bioprinting”, where “time” serves as the fourth dimension combined with 3D bioprinting [7]. Here, “time” refers to the continued development of printed 3D biocompatible materials or live cell structures over time [8]. Three-dimensional bioprinting has brought great hope for enhancing the biological activity of materials to promote autogenous bone regeneration. Our team previously developed a digital 3D bioprinter using bio-printing intelligent manufacturing technology, namely, integrated 3D printing technology with smart materials and cells (PCI3D) [9]. The PCI3D functional module we constructed enhances the biocompatibility of hard materials and provides nutrient channels to support cell proliferation and differentiation, thereby generating a repair function as an intelligent material with spatiotemporal efficacy.

Creating a suitable microenvironment for bone development may be a way to enhance the biological activity of 4D intelligent materials for bone repair. Research has shown that the Wnt signaling pathway is involved in regulating various cellular processes, including cell proliferation, differentiation, migration, polarity, stemness, and lineage plasticity [10,11]. Wnt ligands communicate by binding to surface receptors on the cells, triggering intracellular signal cascades [12]. In canonical Wnt signaling, Wnt ligands bind to Frizzled receptors and low-density lipoprotein receptor-related protein 5 (LRP5) or LRP6, stabilizing β-catenin activation, allowing β-catenin to translocate to the cell nucleus [13]. In the cell nucleus, β-catenin binds with T-cell factor (TCF)/lymphoid enhancer factor (LEF) to activate Wnt signal transcriptional activity [14]. Our previous research found that osteocytes are target cells in bone metabolism regulated by canonical Wnt signaling [15], and the proportion of osteocytes in bone tissue is as high as 90% or more, with an absolute numerical advantage. Therefore, activating osteocytes’ canonical Wnt signaling through 3D printing to provide a developmental microenvironment for bone regeneration holds the potential for regulating bone repair.

Wnt agonists have been widely explored for the treatment of orthopedic-related diseases [16,17]; small-molecule drugs with good stability are particularly useful as candidate drugs [16]. Coghlan et al. developed SB216763 (S33), a highly selective, cell-permeable GSK-3β small molecule inhibitor [18]. In the absence of Wnt ligands, GSK3, adenomatous polyposis coli (APC), Axin, and casein kinase Iα (CKIα) form a complex that phosphorylates β-catenin, which is then ubiquitinated and degraded by the proteasome [13,19]. When GSK3 is inhibited, β-catenin can translocate to the cell nucleus, promoting gene expression through TCF/LEF to activate Wnt signaling [18]. In this study, we stimulated osteocytes with Wnt agonist S33 for 24 h, activating osteocytes’ Wnt signaling to promote osteoblast differentiation in the microenvironment. After stimulating the osteocytes with S33 for 24 h, these osteocytes still facilitate osteoblast differentiation functionality even after the removal of S33. Therefore, short activation of osteocytes Wnt signals followed by the withdrawal of the drug can reduce the risk of abnormal developmental effects or increase certain types of tumors caused by the sustained action of Wnt drugs. In this research, we hypothesize that osteocytes treated with S33 contribute to the formation of a supportive developmental osteogenic microenvironment (DOME). Utilizing a 3D bioprinting system named DOME-PCI3D, a multifunctional 4D intelligent osteogenic module, we aim to establish an advanced developmental system for bone repair.

Our research found that DOME facilitates osteoblast differentiation and the mineralization of ST2 bone marrow stromal cells and promotes bone regeneration and repair. Additionally, DOME can suppress adipogenesis. In addition, DOME also promotes the vascular generation of endothelial cells. Through 3D bioprinting, the 4D intelligent osteogenic module can stably support normal cell growth, and DOME significantly boosts cell proliferation, sustaining osteoblast differentiation for 7–14 days and mineralization for up to 28 days. Therefore, this study indicates that DOME creates a conducive microenvironment for safe bone development, making the 3D bioprinted multifunctional 4D intelligent osteogenic module a promising strategy for bone tissue engineering, particularly in the realms of bone regeneration and repair.

## 2. Materials and Methods

### 2.1. Cells

Human Umbilical Vein Endothelial Cells (HUVEC) were procured from the American Type Culture Collection (ATCC) (Manassas, VA, USA). Additionally, the murine ST2 cell line was obtained from Dr. Steve Teitelbaum at Washington University. The mouse osteocyte-like immortalized cell line MLO-Y4 was provided by Professor Lynda Bonewald from the Indiana University School of Medicine.

### 2.2. Cell Culture

Cell culture reagents α-MEM and DMEM were supplied by Gibco (Beijing, China), while penicillin–streptomycin solution was obtained from Beyotime Biotechnology (Shanghai, China). β-glycerophosphate, ascorbic acid, hydrocortisone, indomethacin, and 3-isobutyl-1-methylxanthine (IBMX) were provided by Sigma (St. Louis, MO, USA). Fetal bovine serum (FBS) was purchased from Biological Industries (Kibbutz Beit Haemek, Israel).

Cells were cultured according to previously described methods [20]. ST2 and MLO-Y4 cells were cultured in complete culture medium containing α-MEM with 10% FBS, 100 U/mL penicillin, and 100 μg/mL streptomycin. MLO-Y4 and ST2 cells were co-cultured at a ratio of 1:4 in 24-well plates for three days.

Osteogenic Induction Medium: Complete culture medium supplemented with 50 µg/mL L-ascorbic acid, 10 mM β-glycerophosphate.

Adipogenic Induction Medium: DMEM complete culture medium supplemented with 0.5 µM hydrocortisone, 0.5 mM IBMX, and 60 µM indomethacin.

HUVECs were cultured in Dulbecco’s Modified Eagle Medium (DMEM) supplemented with 10% FBS, 100 U/mL penicillin, and 100 μg/mL streptomycin. Cells were passaged upon reaching 80–90% confluency using trypsin digestion. All cells were cultured in an incubator at 37 °C with 5% CO_2_.

### 2.3. Activation and Inhibition of Wnt Signaling in Osteocyte MLO-Y4

Chemicals: SB216763 (HY-12012, MedChemExpress, Monmouth Junction, NJ, USA) were dissolved in DMSO (D2650, Sigma) at 60 mM. And iCRT 14 (HY-16665, MedChemExpress, Monmouth Junction, NJ, USA) was dissolved in DMSO at 20 mM concentration.

S33 was used to activate Wnt signaling in MLO-Y4 cells, defining these as DOME. Simultaneously, MLO-Y4 cells treated with DMSO served as the control group. In addition, the Wnt signaling was suppressed by iCRT 14 [21]. To activate Wnt signaling in MLO-Y4 cells, cells were plated in 24-well plates at a density of 1 × 10^4^ cells/well and treated with S33 at concentrations of 0, 5, 10, 20, 30, or 40 µM for 24 h. For specificity in measuring the effect of Wnt signaling in MLO-Y4 cells on osteoblast differentiation, MLO-Y4 cells were treated with S33 at 20 µM for 24 h. At the same time, treatment with iCRT 14 (10 µM) for 24 h inhibits Wnt signaling in MLO-Y4 cells.

### 2.4. Cell Proliferation Analysis

Cell proliferation was analyzed using the CCK8 assay kit (Yeasen Biotechnology, Shanghai, China). MLO-Y4 cells were cultured in 96-well plates (2000 cells/well) and stimulated with various concentrations of S33 for one day. Then, we tested the proliferative activity on the day of withdrawal and the third day. After adding 10 µL of CCK8 solution to each well, cells were further incubated at 37 °C with 5% CO_2_ for 2 h. Absorbance at 450 nm was measured using the Varioskan^TM^ LUX multi-function microplate reader (VL0L00D2, Thermo Fisher Scientific, Waltham, MA, USA). For detecting cell proliferation on PCI3D scaffolds, the above procedure was performed after 1, 4, and 7 days of culture.

### 2.5. Immunofluorescence

Immunofluorescence assays were performed as previously described [20]. MLO-Y4 cells treated with S33 or DMSO were cultured on 24-well plates. After 24 h, cells were washed with Phosphate-Buffered Saline (PBS) fixed with 4% formaldehyde solution at room temperature for 15 min. Following washing, cells were permeabilized with 0.25% Triton X-100 for 30 min and blocked with 5% Albumin Bovine (BSA, 4240GR100, Saiguo Biotechnology, Guangzhou, China) for 30 min. Subsequently, the cells were incubated at 4 °C overnight with a rabbit polyclonal anti-mouse β-catenin antibody (17565-1-AP, Proteintech, Wuhan, China) that was diluted to a 1:100 ratio. Following the wash, the cells were then incubated with Cy3-labeled Goat Anti-Rabbit IgG (H+L) (A0516, Beyotime Biotechnology, Shanghai, China), which was diluted at a 1:200 ratio at room temperature in darkness for a period of 30 min. Following washing, DAPI (C1006, Beyotime Biotechnology, Shanghai, China) was applied for 3 min at room temperature, and after three washes, cells were then observed and imaged using a fluorescence microscope.

### 2.6. Extraction of RNA and Analysis of Gene Expression

Methods for extracting cellular RNA and analyzing gene expression through qPCR were as previously described [22]. To summarize, cells were processed to extract total RNA utilizing Trizol, followed by the synthesis of cDNA using the Evo M-MLV RT Kit. Subsequently, real-time PCR was performed with the aid of the SYBR Green Premix Pro Taq HS qPCR Kit. The Trizol (AG21101), Evo M-MLV RT Kit with gDNA Clean for qPCR (AG11728), and SYBR Green Premix Pro Taq HS qPCR Kit (AG11701) used in the study were all sourced from Accurate Biotechnology, located in Changsha, China.

The primer sequences are detailed in Table 1. In this research, glyceraldehyde-3-phosphate dehydrogenase (*Gapdh*) was utilized as the reference housekeeping gene. The relative expression levels of the mRNA of the target genes in comparison to the housekeeping gene were calculated using the 2^−ΔCt^ method.

### 2.7. AP Staining and Quantitative Analysis

Alkaline phosphatase (AP) staining was performed as previously described [15]. DOME was co-cultured with ST2 cells for three days and subsequently subjected to a washing process using PBS, fixed in 4% formaldehyde for 5 min. Following three consecutive washes with PBS, the cells were stained using the BCIP/NBT Alkaline Phosphatase Color Development Kit (C3206, Beyotime Biotechnology, Shanghai, China) for a period of 30 min. Cells co-cultured on PCI3D scaffolds with ST2 and DOME were stained for 4 h after 7 and 14 days of culture. Images of the samples were captured using a digital camera.

For the analysis of AP biochemical activity, cells were initially washed with PBS and then treated with 300 µL of 10 mM Tris/HCl solution (pH 7.4). Cells were then collected and lysed using ultrasonication. Following centrifugation at a speed of 12,000 rpm for a duration of 3 min, the supernatant was then carefully extracted. The supernatant was then used for detection using alkaline phosphatase Assay Kit (P0321S, Beyotime Biotechnology, Shanghai, China). The absorbance was recorded at a wavelength of 405 nm using the Varioskan^TM^ LUX multi-function microplate reader.

### 2.8. Staining with Alizarin Red S and Its Quantitative Assessment

In accordance with methodologies reported in prior research [23], after initially culturing DOME and ST2 cells in the complete culture medium for 3 days under 2D conditions or 7 days under 3D conditions, the cells were subsequently cultured in the osteogenic induction medium for an additional 14 days (2D) or 21 days (3D scaffolds). During these periods, half of the medium was systematically replenished every two days. Specimens underwent a staining process for 30 min using a 1% Alizarin Red S solution (Solarbao Biotechnology, Beijing, China), followed by photographic documentation. Following this, the sample was incubated with 10% cetylpyridinium chloride solution for a duration of 1 h. The absorbance of this solution at 562 nm was then quantified using the Varioskan^TM^ LUX multi-function microplate reader.

### 2.9. Adipogenic Differentiation and Oil Red O Staining

Briefly, ST2 cells were co-cultured with DOME for three days, after which the medium was replaced with the adipogenic culture medium and incubated for seven days. ST2 cells were stained using a modified Oil Red O staining kit (C0157S, Beyotime Biotechnology, Shanghai, China) and detected by inverted fluorescence microscopy (Nikon, Tokyo, Japan).

### 2.10. Cell Migration Analysis

In accordance with the experimental method previously described in reference [20], we added MLO-Y4 to a 24-well plate and treated the cells with S33 for one day, removed the drug, and cultured it for two more days. Finally, a total of 10,000 HUVEC cells were seeded into the upper chamber of a Transwell apparatus (Corning, New York, NY, USA). Following seeding, the cells were subjected to an incubation period of 24 h at 37 °C in an atmosphere containing 5% CO_2_. Cells remaining inside the Transwell were carefully removed, and the cells that had migrated to the lower surface were stained using crystal violet for a duration of 5 to 10 min. Post-migration cell imaging was conducted with a microscope, and quantitative analysis was performed using the ImagJ software (64-bit, v1.46).

### 2.11. HUVEC Tube Formation Assay

Following the experimental protocol described in reference [20], the function of DOME on angiogenesis of HUVECs was tested. Briefly, a mixture comprising 120,000 HUVEC cells and 30,000 either DOME or control cells was prepared. The mixture was then seeded onto the surface of Matrigel (Corning, NY, USA) and was pre-coated in a 24-well plate. The cells were cultured for six hours. Capillary-like structures were observed and photographed under a microscope, and the HUVEC tube networks were quantitatively analyzed using ImageJ.

### 2.12. Live/Dead Assay

Cell viability for those cultured on the PCI3D module was assessed using the Calcein/PI Cell Viability/Cytotoxicity Assay Kit (C2015M, Beyotime Biotechnology, Shanghai, China), with evaluations conducted after 1, 4, and 7 days. That is, 1 mL of assay buffer was incubated in a staining mixture of 1 µL Calcein AM (1000×) and 1 µL Propidium Iodide (PI) (1000×). Calcein AM is used to stain live cells, which then exhibit green fluorescence. Conversely, PI is employed for staining dead cells, resulting in red fluorescence. After incubating the samples for 0.5 h at 37 °C in an incubator, they were promptly imaged using an inverted fluorescence microscope from Leica, based in Wetzlar, Germany. The quantification of cell viability was conducted utilizing the ImageJ software (64-bit, v1.46).

### 2.13. Integrated 3D Printing

As we previously reported [9], the creation of a polycaprolactone (PCL) and cell-incorporated 3D (PCI3D) module was achieved using a dual-head biomanufacturing 3D printer. This printer utilized one head for extruding PCL, providing rigid structural support, and another head dedicated to printing a cell-laden hydrogel, which contained a mix of MLO-Y4 and ST2 cells in a 1:4 ratio aimed at facilitating osteogenic differentiation. The lyophilized Gelatin Methacryloyl (GelMA, SP-BI-G01-2, Sunp Biotech, Beijing, China) was thoroughly dissolved in α-MEM, resulting in a 20% (*w*/*w*) GelMA solution. This solution also included 0.5% (*w*/*v*) of a photoinitiator, LAP (SP-BI-C02-1, Sunp Biotech, Beijing, China). The prepared solution was then preserved at a temperature of 4 °C for future applications. The PCL nozzle was pre-heated at 95 °C for 30 min to reach optimal printing temperature; molten PCL was then printed with a 400 µm diameter and 1100 µm spacing at a speed of 2 mm/s. Concurrently, 2 × 10^5^ MLO-Y4 cells and 8 × 10^5^ ST2 cells were mixed with an equal volume of the aforementioned GelMA solution. The GelMA solution, infused with cells, was extruded between strands of PCL at a print speed of 5 mm/s. The process was conducted with a cell bundle diameter set to 300 µm and maintained a spacing of 500 µm between the prints [24]. After the printing of each layer, the GelMA hydrogel underwent crosslinking under 405 nm blue light for 10 s. This step was then followed by the vertical printing of the subsequent layer. Upon the completion of three layers’ printing, a 3D scaffold of PCL and cells (PCI3D) was successfully fabricated. The PCI3D module was then placed in the complete culture medium and incubated in a 37 °C, 5% CO_2_ incubator for subsequent experiments.

### 2.14. Statistical Analysis

The statistical analyses were performed utilizing the GraphPad Prism 8.0.1 software. Every experiment was independently replicated a minimum of three times. The results are expressed as the mean ± SD (standard deviation). The Student’s *t*-test was utilized for the analysis of variations between the two groups. One-way ANOVA was used for comparing differences across multiple groups, and two-way ANOVA was used for evaluating the effects of two independent variables. A *p*-value below 0.05 was deemed to indicate statistical significance.

## 3. Results

### 3.1. Activation of Wnt Signaling in Osteocytes by S33

The MLO-Y4 osteocyte cell line was stimulated using the Wnt agonists S33, which was employed to initiate the canonical Wnt signaling pathway within these osteocytes. To ascertain the ideal concentration of S33 required to stimulate Wnt signaling in MLO-Y4 cells, we administered S33 to these cells at various concentrations: 0, 5, 10, 20, 30, and 40 μΜ. After one day of treatment with S33, we tested the proliferative activity on the day of withdrawal and the third day. The proliferative activity of MLO-Y4 cells showed no notable changes at concentrations of S33 ranging from 0 to 30 µM. However, a concentration of 40 µM significantly inhibited the proliferation of osteocytes at both time points (Figure 1A). Subsequently, through quantitative real-time fluorescence assays, we confirmed that S33 at the concentration of S33 between 20 μM and 40 μM enhanced the expression of Wnt signaling pathway target genes (e.g., *Lef1* and *Axin2*) in MLO-Y4 cells (Figure 1B). Consequently, we chose 20 μM as the working concentration of S33 for this research. To further confirm S33’s activation of the Wnt signaling pathway in osteocytes at the protein level, immunofluorescence experiments revealed that S33 at 20 μM stimulated the nuclear translocation of β-catenin in MLO-Y4 cells, thereby activating Wnt signaling (Figure 1C). We evaluated the activation of Wnt signaling in osteocytes by S33 over 1, 3, and 5 days post-withdrawal. Our observations revealed a significant enhancement in Wnt signaling one day after S33 treatment. On the 3rd day, despite a slight decline, Wnt signaling remained significantly elevated compared to the DMSO (control) group. By the 5th day, the level of Wnt signaling in osteocytes reduced to levels not significantly different from those observed in the DMSO group (Figure 1D).

### 3.2. The Effect of S33-Stimulated Osteocytes on Osteoblast Differentiation

After treating MLO-Y4 cells with S33 at concentrations of 0, 5, 10, 20, 30, or 40 μM for 24 h and then replacing the medium, we co-cultured the osteocytes with ST2 cells for three days. We found that only at a concentration of 20 μM S33 significantly enhances alkaline phosphatase (AP) staining in ST2 cells (Figure 2A). In our co-culture osteogenic differentiation tests, we discovered that at 5 µM of S33, the alkaline phosphatase (AP) activity and osteogenic gene expression were similar to the control group. At a concentration of 10 µM, AP activity and *Alpl* gene expression increased, but *co1a1* and *sp7* showed no significant change. A dose of 20 µM significantly enhanced AP activity and upregulated the expression of *Alpl*, *Co1a1*, and *Sp7* genes. At 30 µM, both AP staining and the expression of *Alpl* and *Co1a1* were significantly higher than in the control group, with no change in *Sp7* expression. With 40 µM, AP activity and *Alpl* levels were higher than in the control group. (Figure 2B,C). Therefore, we decisively chose to use 20 μM of S33 to activate Wnt signaling in osteocytes and examine its effects on osteoblast differentiation. Alizarin Red S staining indicated that S33 markedly enhanced the formation of mineralized nodules (Figure 2D). Compared to the control group, calcium deposition in the S33-treated MLO-Y4 osteocyte line increased by 1.5-fold (Figure 2E). These data suggest that S33-activated osteocytes create a developmental osteogenic microenvironment conducive to osteoblast differentiation, hereafter referred to as DOME.

### 3.3. Inhibition of Adipogenic Differentiation in ST2 Cells by DOME

Adipogenesis refers to the differentiation of mesenchymal stem cells into fully developed, lipid-rich adipocytes [25,26]. Thus, we examined if S33-activated canonical Wnt signaling in osteocytes impacts adipogenic differentiation. Microscopic examination, after seven days of culturing ST2 cells with DOME in an adipogenic medium, revealed diminished lipid droplet formation compared to the control group (Figure 3A). Moreover, DOME inhibited the expression of *Cebpa* and *Pparg* in ST2 cells, along with a decrease in the expression of the mature adipocyte marker gene *Fabp4* (Figure 3B). These findings indicate that DOME can inhibit adipogenic differentiation in ST2 cells.

### 3.4. Promotion of Angiogenesis by DOME

Research indicates that angiogenesis significantly enhances bone repair [27]. Thus, we explored the potential of DOME to induce angiogenesis. We observed that DOME promoted cell migration in HUVECs (Figure 4A). Relative to the control group, there was a 1.75-fold increase in the number of migrating cells (Figure 4B). Furthermore, we assessed tube formation in HUVECs co-cultured with DOME. In comparison to the control group, DOME boosted the formation of vascular tubules (Figure 4C). The count of formed nodes, total length, and branching lengths rose by 1.8-fold, 1.5-fold, and 1.4-fold, respectively, against the control group (Figure 4D). Additionally, compared to the control, S33 was found to elevate the expression of the angiogenesis gene *Vegfa* in MLO-Y4 cells (Figure 4E). These results suggest that DOME enhances angiogenesis.

### 3.5. The Key Role of Wnt/β-Catenin Signaling in DOME for Promoting Osteoblast Differentiation and Inhibiting Adipogenic Differentiation

To verify whether DOME’s enhancement of osteoblast differentiation was a result of canonical Wnt/β-catenin signaling, we employed the canonical Wnt signaling inhibitor iCRT 14. iCRT 14 directly interferes with the interaction between β-catenin and the nuclear transcription factor TCF4, consequently inhibiting Wnt signaling. RT-qPCR results indicated that 10 μM of iCRT 14 effectively reduced the expression of the Wnt target genes *Lef1* and *Axin2*, which were upregulated by DOME (Figure 5A). Following this, the osteoblast differentiation enhanced by DOME was diminished by iCRT 14 (Figure 5B–D). The results demonstrated that iCRT 14 impeded the increase and activity of AP (Figure 5B,C), and the co-cultured qPCR results also indicated that iCRT 14 downregulated the osteogenic marker genes (*Alpl*, *Col1α1*, *Sp7*) upregulated by DOME (Figure 5D). Meanwhile, iCRT 14 inhibited the increase in mineralization nodules promoted by DOME (Figure 5E,F). Thus, DOME promotes osteogenic differentiation of ST2 cells through canonical Wnt signaling. At the same time, Oil Red O staining revealed that adipogenesis inhibited by DOME was restored to control levels by iCRT 14 (Figure 5G). Gene detection also found that the addition of iCRT 14 upregulated the expression of *Pparg* and *Cebpa*, and mature adipocyte gene *Fabp4* expression returned to control levels (Figure 5H). Therefore, DOME inhibits the adipogenic differentiation of ST2 cells. These data demonstrate the key role of canonical Wnt signaling activated by DOME in osteogenic differentiation.

### 3.6. DOME Promotes Cell Proliferation in the PCI3D Scaffold

To improve two-dimensional cell culture and better mimic the in vivo bone environment, we previously developed a PCL-integrated 3D printing scaffold as an in vitro model (PCI3D) (Figure 6A), supporting cell–cell interactions to simulate the in vivo bone microenvironment. We conducted a co-culture of PCI3D modules, which contained MLO-Y4 and ST2 cells treated with S33, over periods of 1, 4, and 7 days. Through the application of Calcein AM and PI staining, we noted a substantial number of live cells adhering to the 3D scaffold in both the DOME and control groups, with the number of live cells progressively increasing over time (Figure 6B). The 4D intelligent osteogenic module (DOME-PCI3D system) demonstrated a high cell survival rate, with 91.2% of cells remaining viable after 7 days in the 3D environment (Figure 6C). These results indicate that cells cultured in the PCI3D scaffold exhibit high viability. Through CCK8 proliferation assays, we found that cell proliferation capacity in the PCI3D modules linearly increased over 7 days (Figure 6D). Compared to the control group, the DOME group showed higher proliferation activity on days 4 and 7. These findings imply that the PCI3D module offers a conducive environment for cell survival and proliferation and that DOME further enhances cell proliferation.

### 3.7. Promotion of Osteoblast Differentiation and Mineralization in the PCI3D Module by DOME

After establishing the advantageous conditions offered by the PCI3D functional module for cell survival and proliferation, we proceeded to investigate the osteogenic differentiation of ST2 cells within the PCI3D module. Upon culturing DOME-ST2 and MLO-Y4-DMSO-ST2 cells in the PCI3D module for both 7 and 14 days, we observed more intense AP staining in the DOME-ST2 group compared to the control group (Figure 7A). Quantitative analysis further revealed that DOME significantly enhances the activity of alkaline phosphatase in ST2 cells within the PCI3D module (Figure 7B). Subsequently, through gene detection, we found significant increases in the expressions of osteogenic marker genes *Alpl*, *Col1α1*, and *Sp7* in the DOME-ST2 group within the PCI3D module (Figure 7C). Therefore, DOME promotes osteogenic differentiation in the PCI3D module.

Following this, we examined the effect of DOME on mineralization in the PCI3D module. The PCI3D module was initially cultured in the complete medium for 7 days, followed by a switch to an osteogenic induction medium for 21 days. Alizarin Red S staining and quantitative analysis revealed that the DOME module formed larger and denser calcium nodules (Figure 7D). Moreover, the degree of mineralization was augmented by approximately 1.8-fold (Figure 7E). These confirm the positive role of DOME in the mineralization of ST2 cells.

## 4. Discussion

The 3D bioprinting to construct a developmental functional osteogenic microenvironment is at the forefront of bone tissue engineering research. We previously discovered that activating Wnt signaling in osteocytes has a dual effect on osteogenesis and osteoclastogenesis [15]. Therefore, we suggest that activating osteocyte Wnt signaling may be applied to promote bone repair. Here, we found that the small-molecule drug S33 can activate osteocyte Wnt signaling. Surprisingly, osteocytes treated with S33 at 24 h were able to maintain Wnt signaling for 3 days after removing medication. Osteocytes treated with S33 create a developmental osteogenic microenvironment (DOME). DOME promotes ST2 cell differentiation into osteoblasts and mineralization, inhibits adipocyte generation, and promotes vascularization. The addition of the Wnt signaling inhibitor iCRT 14 could suppress the osteogenic effect induced by DOME. Based on this, we successfully designed and tested a novel multifunctional 4D intelligent osteogenic module for bone tissue engineering in bone repair. Experimental results show that in this 4D intelligent osteogenic module, DOME can stably support ST2 cells’ growth and promote ST2 cells’ osteoblast differentiation and mineralization. This study found that the 4D intelligent osteogenic module has excellent osteogenic effects, making it a promising candidate for bone tissue engineering.

A highlight of this study is the identification of a safe, usable small-molecule drug for activating Wnt signaling in osteocytes. In bone tissue engineering, small-molecule drugs offer unique advantages, including precise control over drug dosage, convenient modulation of treatment time windows, and a lower risk of side effects [28,29]. GSK-3 inhibitors, by blocking the activity of GSK-3 and preventing β-catenin degradation, activate the Wnt/β-catenin signaling pathway [18]. In other studies, GSK-3 inhibitors were shown to play a significant role in various pathological conditions, including neurological diseases, diabetes, and cancer [30,31,32,33]. For instance, in neuro-degenerative disease research, GSK-3 inhibitors promote neuronal survival and regeneration by activating Wnt signaling [34]. Our study demonstrates that the GSK-3 inhibitor small-molecule drug S33 is a safe osteogenic drug. S33 not only activates osteocyte Wnt signaling in a short induction but also maintains robust osteogenic activity post-drug withdrawal. Therefore, one of the highlights of this study is that osteocytes can maintain osteogenic functions after brief stimulation with the small-molecule drug S33, a method that foregoes the use of transgenic induction and offers increased safety.

This study also successfully established and validated DOME, a safe developmental osteogenic microenvironment. DOME contains no drug components, and the short stimulation with S33 followed by timely drug withdrawal avoids the indefinite activation of Wnt signaling in other cells [35], making it safe for application in bone tissue engineering. This has significant implications for the treatment of bone repair in patients.

DOME, through Wnt/β-catenin signaling, promotes the osteoblast differentiation of bone marrow stromal cells while inhibiting adipocyte differentiation. Recognized as multipotent progenitor cells, bone marrow stromal cells can differentiate into adipocytes, osteoblasts, or chondrocytes under suitable conditions. This study found that DOME promotes osteoblast differentiation of ST2 cells and inhibits adipocyte differentiation. Here, we discovered that the canonical Wnt signaling inhibitor iCRT 14 reduced the osteogenic potential of DOME and reversed its inhibitory effect on adipocyte differentiation, thus confirming that DOME promotes osteoblast differentiation and inhibits adipocyte in ST2 cells via the activation of Wnt/β-catenin signaling in osteocytes.

Additionally, this study found that *vegfa* was elevated in DOME and effectively promotes angiogenesis, a vital component of bone repair. Blood vessels not only provide essential nutrients and oxygen but also enhance the functionality of osteoblasts and other cell types, thereby accelerating the repair process of bone defects [27,36]. Another crucial aspect of bone tissue engineering is the correct dosage of growth factors [37]. We observed that the appropriate treatment dosage is crucial to avoid the malformations and leaky vessels potentially caused by high-dose treatments with VEGF and FGF [27,38]. Therefore, our treatment strategy focuses on using the minimal effective dose to ensure long-term stability and safety. The short therapeutic window of our drug administration effectively avoids adverse reactions due to excessive drug dosage.

The biggest highlight of this study is the pioneering advancement in the development of biomaterials for bone repair [39]. Building upon the previously established physicochemical and bioactive microenvironment PCI3D functional modules [9], we utilized 3D printing to construct bone repair functional modules with metabolic functions. This approach effectively overcomes the challenges of shape conformity and limited biological activity, which have long been bottlenecks in orthopedic material design. Currently, 3D printing used in bone repair mainly involves material shape deformation or immune regulation to promote bone repair [40,41]. Distinctly, our strategy establishes a supportive developmental osteogenic microenvironment for bone development, thereby directly facilitating bone repair and regeneration. Compared to traditional 2D culture systems, the PCI3D system creates a biomimetic environment that more closely mirrors the physical structure of natural bone tissue for DOME cells. Employing rigid material scaffolds not only replicates bone tissue’s physical architecture but also promotes three-dimensional cell interactions. Such interactions significantly improve cellular functions, including the capacity of bone marrow stromal cells for differentiation into osteoblasts and the maintenance of bone tissue’s unique microenvironment. In contrast, the 2D culture system confines cells to growth and interactions on a planar surface. This restriction impedes their spatial behavior and functional expression, markedly diverging from the in vivo multidimensional complex environment. Consequently, the 4D intelligent osteogenic module furnishes an optimized and efficacious microenvironment for the osteogenic development of DOME cells compared to the 2D system. This advancement not only enriches our comprehension of bone tissue engineering principles but also sets the stage for future clinical applications.

In the future, we aim to further explore the controllable regulatory mechanism of bone marrow stromal cell-directed osteogenic differentiation microenvironment and its potential applications in treating bone diseases such as osteoporosis and bone defects.

## 5. Conclusions

The small-molecule drug SB216763, in low doses, stimulates osteocytes, activating Wnt signaling and, thereby, forming a developmental osteogenic microenvironment. DOME induces osteoblast differentiation and the mineralization of bone marrow stromal cells while inhibiting adipogenic differentiation and enhancing endothelial cell-mediated angiogenesis. Additionally, the multifunctional 4D intelligent osteogenic module constructed by 3D bioprinting provides a developmental osteogenic microenvironment to promote osteoblast differentiation. This multifunctional 4D intelligent osteogenic module improves the shape matching of bone repair and enhances the biological activity of bone repair, thus having vast application and translational value for bone regeneration.

## Figures and Tables

**Figure 1 biomolecules-14-00354-f001:**
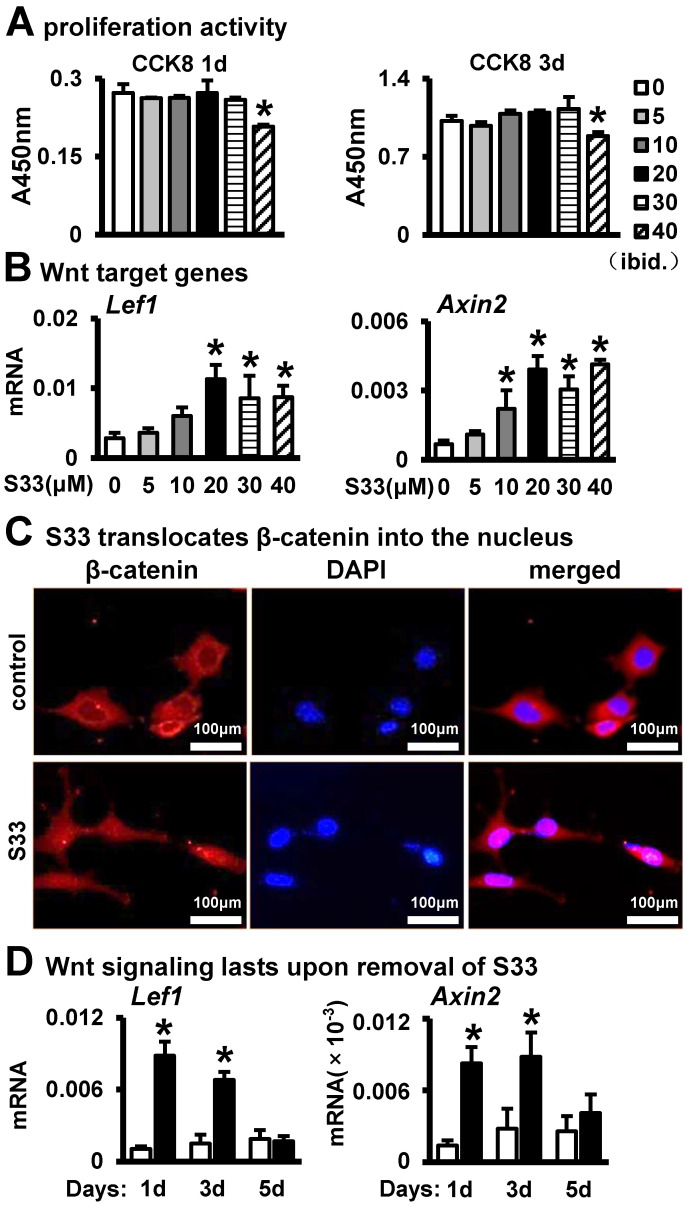
S33 activates the Wnt signal of MLO-Y4 cells. (**A**) CCK8 assay of proliferative activity of MLO-Y4 cells on the day of withdrawal and the third day after treatment with S33 (0, 5, 10, 20, 30 or 40 µM) for one day. The * indicates *p* < 0.05 versus 0 µM group by one-way ANOVA; n = 3. (**B**) Expression of Wnt signaling target genes in MLO-Y4 cells treated with S33 for 24 h, as detected by qPCR. Level of mRNA versus housekeeping gene *Gapdh*. * *p* < 0.05 versus 0 µM group by one-way ANOVA, n = 3. (**C**) Immunofluorescence staining of β-catenin in MLO-Y4 cells treated with DMSO (control) or S33 (20 µM). Scale bar = 100 μm. (**D**) Expression of Wnt signaling target genes in MLO-Y4 cells after removing S33 (20 µM), as detected by qPCR. *, *p* < 0.05, compared with the control group by *t*-test, n = 3.

**Figure 2 biomolecules-14-00354-f002:**
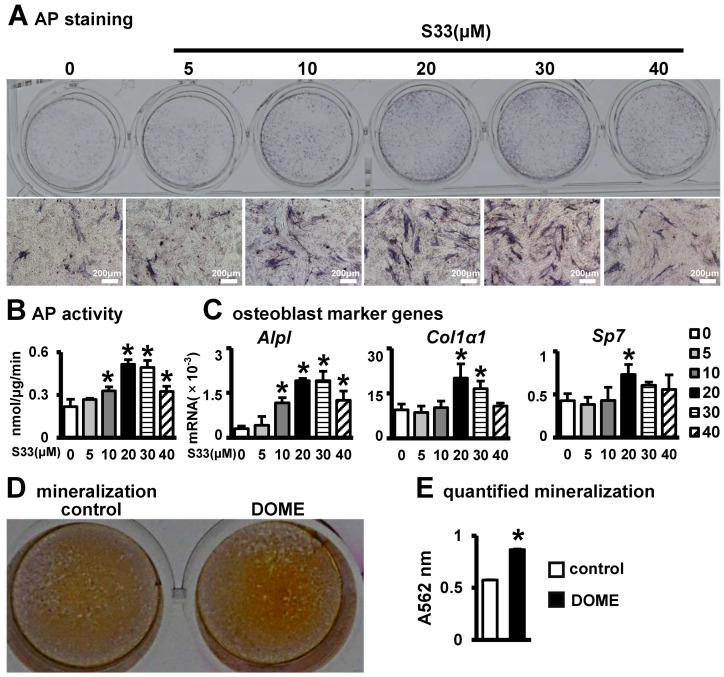
DOME-induced osteogenic differentiation of ST2 cells. (**A**,**B**) AP staining and AP biochemical activity. MLO-Y4 was treated with 0, 5, 10, 20, 30, or 40 µM of S33 for 24 h, then co-cultured with ST2 cells for 3 days. Scale bar = 200 µm. The * indicates *p* < 0.05 versus control group by one-way ANOVA; n = 3. (**C**) Expression of osteoblast marker genes as measured by qPCR. The * indicates *p* < 0.05 versus control group by one-way ANOVA; n = 3. (**D**) Alizarin Red S staining. ST2 cells were co-cultured with DOME in growth medium for 3 days and incubated in osteogenic medium for another 14 days for the mineralization assay. (**E**) Quantitative analysis of mineralization. *, *p* < 0.05, compared with the control group by *t*-test, n = 3.

**Figure 3 biomolecules-14-00354-f003:**
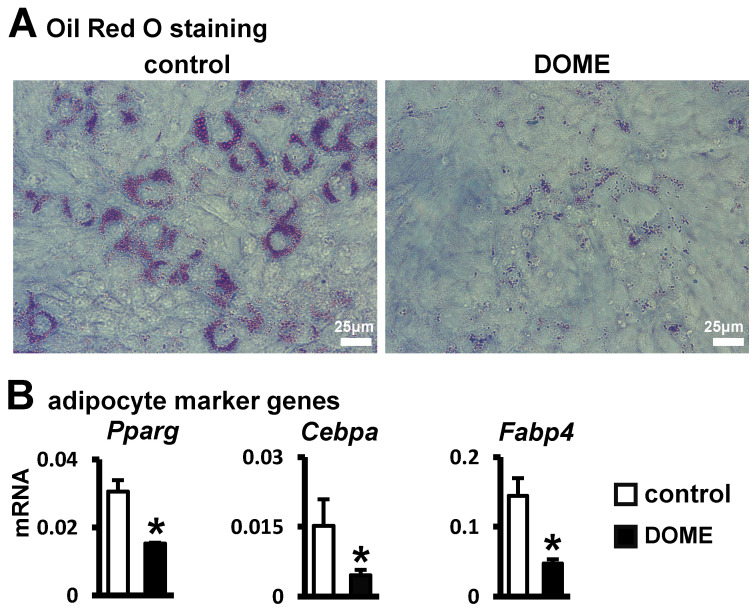
DOME-mediated inhibition of adipogenic differentiation in ST2 cells. (**A**) Oil Red O staining. ST2 cells were cultured with DOME in Adipogenic Induction Medium for 3 days, and then induced with adipogenic medium for 7 days. Scale bar = 25 µm. (**B**) qPCR assay for the detection of adipocyte marker genes. *, *p* < 0.05, compared with the control group by *t*-test, n = 3.

**Figure 4 biomolecules-14-00354-f004:**
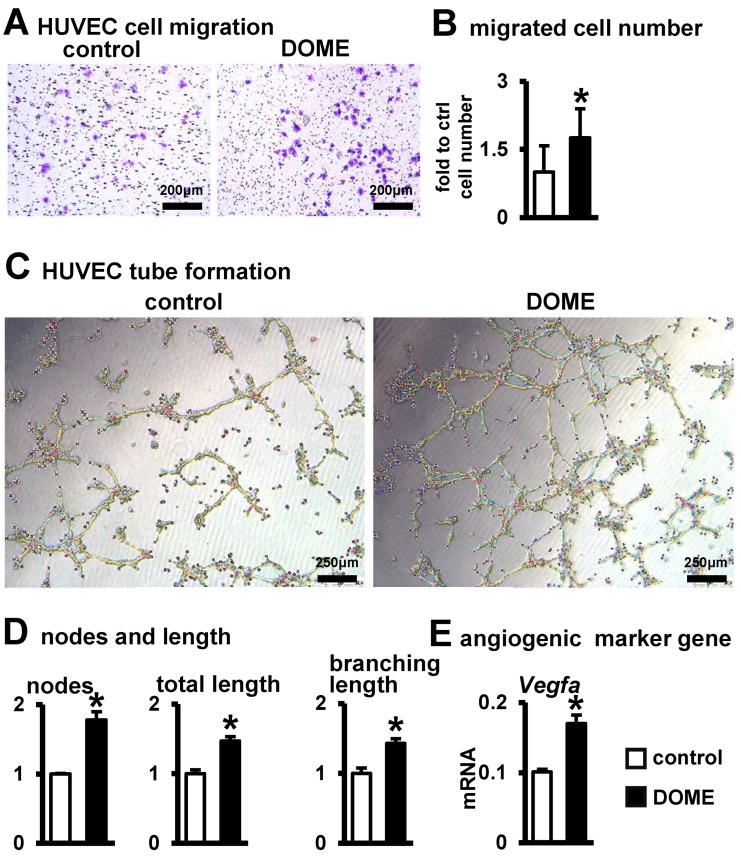
DOME promotes angiogenesis. (**A**) Images of HUVECs migration assay. MLO-Y4 in inoculated 24-well plates were treated with S33 for one day; then, the drug was removed and cultured for two more days. Finally, HUVEC cells were seeded into the upper chamber of the Transwell, and Crystalline violet staining was performed after 24 h of incubation. Scale bar = 200 µm. (**B**) Quantification of migrated cells. (**C**) Images of vascular tubules forming. HUVECs were inoculated with DOME at a ratio of 1:4 onto 200 µL Matrigel in pre-coated 24-well plates and cultured for 6 h. Scale bar = 250 µm. (**D**) Calculation of formed nodes, total length, and branch length. (**E**) Expression of angiogenic marker Vegfa. *, *p* < 0.05 by *t*-test, compared with the control group, n = 3.

**Figure 5 biomolecules-14-00354-f005:**
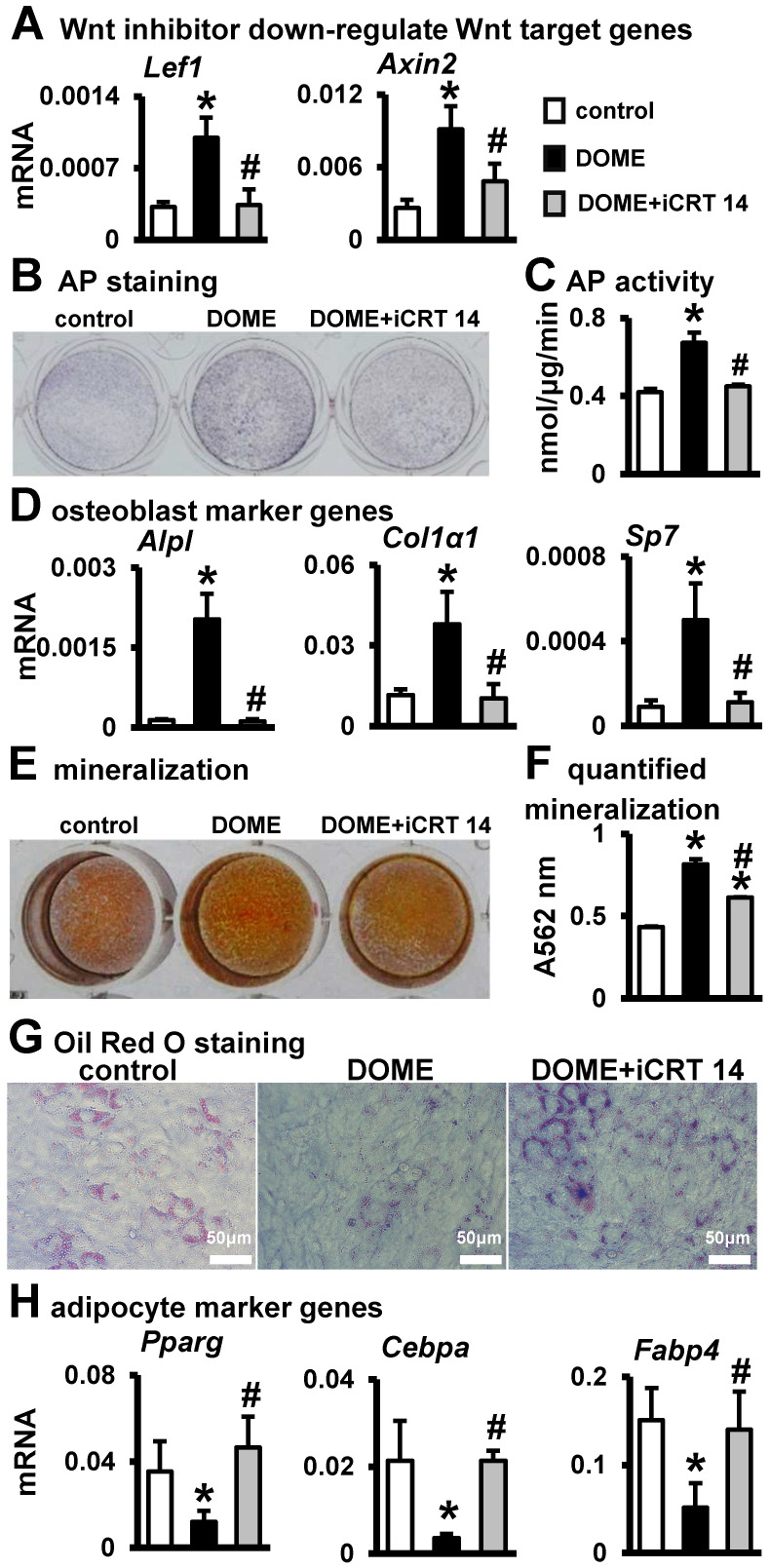
DOME enhances osteogenic differentiation and inhibits adipogenic differentiation via the Wnt/β-catenin signaling pathway. (**A**) Expression of Wnt target genes in MLO-Y4. The DOME was treated with or without Wnt inhibitor iCRT 14 (10 μM) for 24 h. (**B**) AP staining. DOME treated with iCRT14 was co-cultured with ST2 cells for 3 days. (**C**) AP biochemical activity assay. (**D**) Expression of osteoblast marker genes. (**E**) Alizarin Red S staining. (**F**) Mineralization quantification assay. (**G**) Oil Red O staining. Scale bar = 50 µm. (**H**) Expression of adipogenic marker genes. *, *p* < 0.05, compared with the control group. #, *p* < 0.05, compared with the DOME group by one-way ANOVA, n = 3.

**Figure 6 biomolecules-14-00354-f006:**
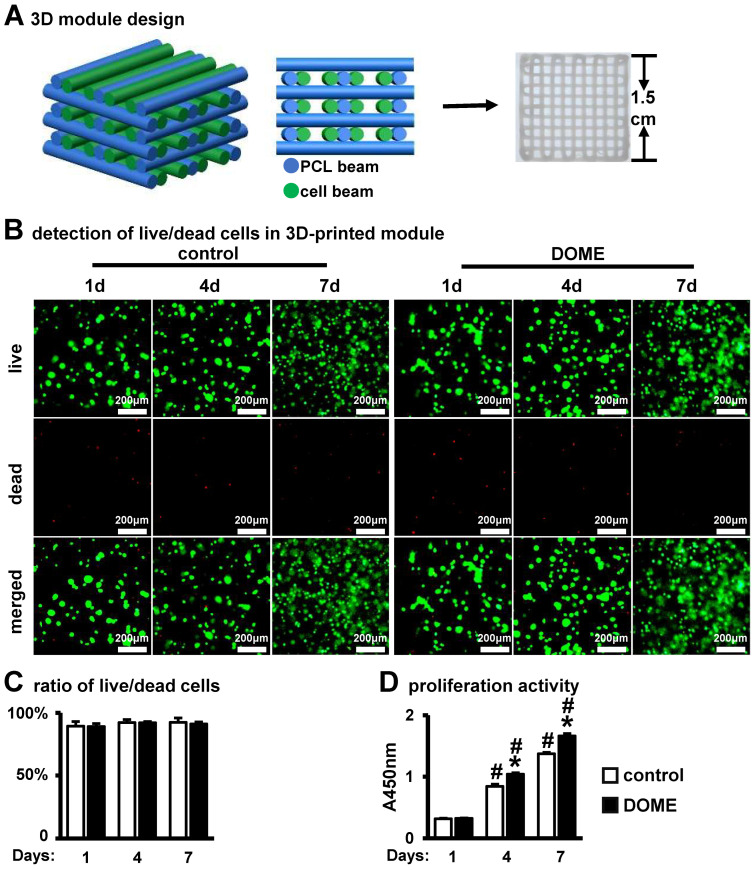
DOME promotes cell proliferation in PCI3D modules without affecting cell survival. (**A**) Schematic diagram (**left**) and photo (**right**) of the PCI3D scaffold. (**B**) Detection of live/dead cells at 1, 4, and 7 days in PCI3D scaffold, scale bar = 200 μm. (**C**) Live/dead cell ratio. (**D**) CCK8 assay for cell proliferation activity. *, *p* < 0.05, compared with the control group, n = 3. #, *p* < 0.05, compared with the Day 1, by two-way ANOVA. n = 3.

**Figure 7 biomolecules-14-00354-f007:**
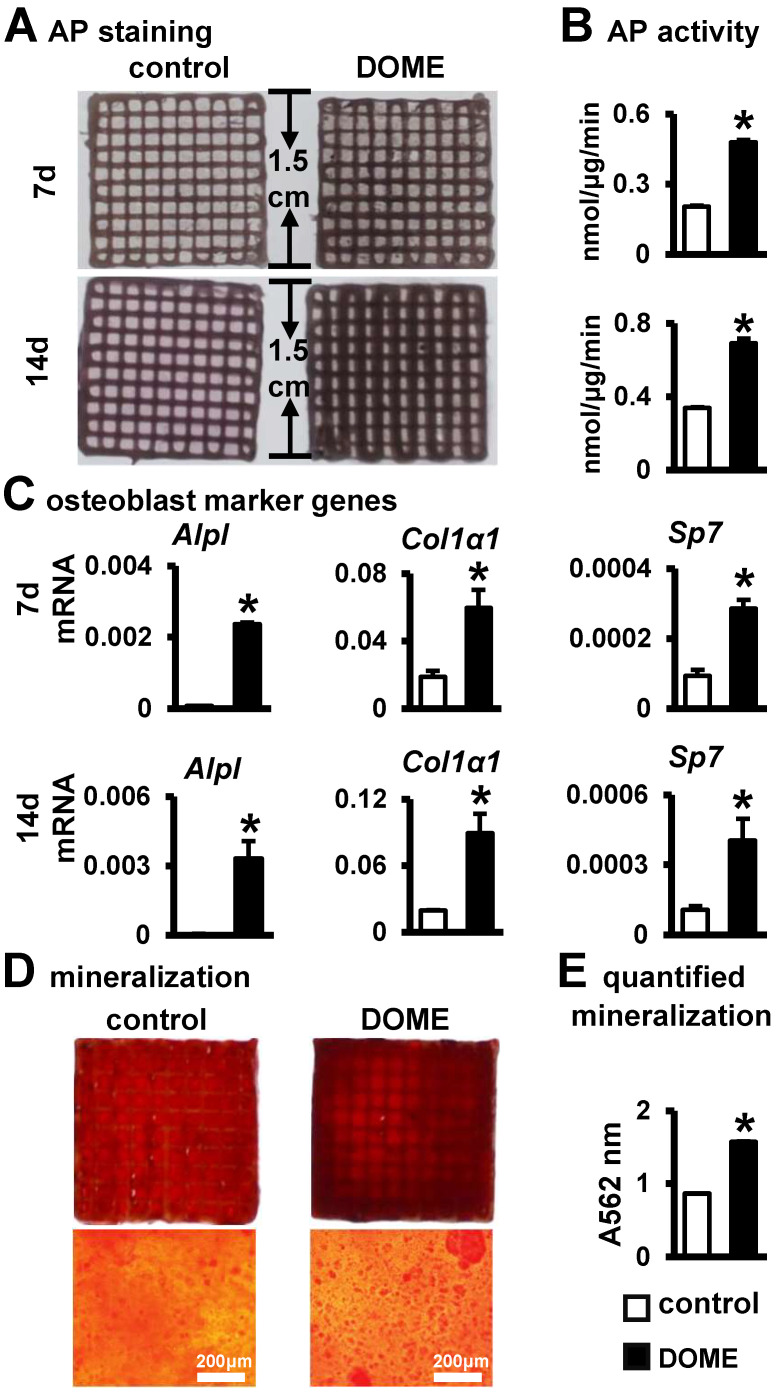
DOME promotes osteogenic differentiation in PCI3D modules. (**A**) AP staining of PCI3D modules cultured for 7 and 14 days. (**B**) AP biochemical activity assay. (**C**) Expression of osteoblast marker genes. (**D**) Alizarin Red S staining images of PCI3D modules under camera and microscope, scale = 200 μm. The modules were cultured in basic medium for 7 days and then cultured in an osteogenic medium for 21 days. (**E**) Mineralization quantification assay. *, *p* < 0.05, compared with the control group by *t*-test, n = 3.

**Table 1 biomolecules-14-00354-t001:** Sequences of primers used for qRT-PCR.

Primer	Forward	Reverse
*Gapdh*	GCACAGTCAAGGCCGAGAAT	GCCTTCTCCATGGTGGTGAA
*Lef1*	CCTACAGCGACGAGCACTTTT	CCTTGCTTGGAGTTGACATCTG
*Axin2*	TGAGCGGCAGAGCAAGTCCAA	GGCAGACTCCAATGGGTAGCT
*Alpl*	TTCGCTATCTGCCTTGCCTG	AGTCTGTGTCTTGCCTGCC
*Col1α1*	TCAACCCCGTCTACTTCCCT	TTCAACAGTCCAAGAACCCCAT
*Sp7*	GCCCCCTGGTGTTCTTCATT	CTTCCCCCTTCTTGGCACTC
*Pparg*	AGCCCTTTACCACAGTTGATTTCTCC	GCAGGTTCTACTTTGATCGCACTTTG
*Cebpa*	TGGACAAGAACAGCAACGAG	TCACTGGTCAACTCCAGCAC
*Fabp4*	GACGACAGGAAGGTGAAGAGCATC	GGAAGTCACGCCTTTCATAACACATTC
*Vegfa*	TTCGAGGAGCACTTTGGGTC	GTGGGTGGGTGTGTCTACAG

## Data Availability

The data presented in this study are available upon request from the corresponding author.

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
