# Peer review of "The Osteocyte with SB216763-Activated Canonical Wnt Signaling Constructs a Multifunctional 4D Intelligent Osteogenic Module"

_biomolecules, 2024, doi:10.3390/biom14030354_

Round 1

Reviewer 1 Report

Comments and Suggestions for Authors

This paper is an investigation to construct a developmental osteogenic microenvironment (DOME) through 4D bioprinting of activated osteocytes and to evaluate its osteogenic differentiation potential for bone marrow stromal cells. Following points should be addressed before paper re-submission.

1) Figure 1: The results of osteocytes activated by adding S33 of more than 20 uM should be added to optimize the best activation condition.

2) Figures 2-5: When the assays were performed should be added in the Figure legends for readers’ better understanding.

3) Figure 2-4: The assays were performed under co-culture condition. I am curious how osteocyte and bone marrow stromal cells were discriminated in this condition.

4) Figure 2C: The reason why the expression of Cola1 gene temporally decreased when osteocytes were cultured with 10 uM of S33

5) The data on how long the activation of osteocytes with S33 maintained should be added.

6) What biomolecules from activated osteocytes promoted osteogenic differentiation for bone marrow stromal cells or angiogenesis should be elucidated.

7) The feasibility of DOME in 3D culture system compared with that in 2D culture system should be discussed based on the results obtained in this paper.

Author Response

Dear Editors and Reviewers,

We are writing to submit the revised version of our manuscript ID biomolecules-2864044, We are grateful for the reviewers' constructive feedback and have addressed each point as follows:

In response to the comments received:

  • Modifications addressing Reviewer #1's feedback are marked in orange.
  • Adjustments made in light of Reviewer #2's suggestions are marked in blue.
  • Revisions pertinent to concerns shared by both reviewers are marked in plum.
  • We have also initiated several clarifications and refinements, marked in purple, to further improve our manuscript.

Significant changes include:

  • Revise the title based on the reviewer #2's comments
  • Adding experimental results for drug concentrations above 20 μM, and updating Figures 1A, 1B, and 2A-C in line with Reviewer #1's suggestions.
  • Modifying Figure 6D as advised by Reviewer #2. The conclusion remains unchanged.
  • Expanding the discussion on the effect of 3D culture on DOME cells.
  • Correction of Figure 5F, and highlighting significant differences between the DOME+iCRT14 group and the DOME group.
  • Re-uploading original images for Figures 3A and 5G to improve clarity without altering the content.

These revisions have not only addressed the reviewers’ concerns but also enriched our manuscript. Attached are the point-by-point responses to the reviewer’s comments and the revised manuscript with changes highlighted.

We believe these revisions make a substantial contribution to the manuscript and hope it now meets the journal’s standards for publication. We welcome any further suggestions and are committed to collaborating with the journal and reviewers to ensure our work meets the highest quality standards.

Thank you for considering our revised manuscript. We look forward to your feedback.

Best regards,

Xiaolin Tu, Jinling Zhang, and Ying Zhang

Reviewer 2 Report

Comments and Suggestions for Authors

Comments on the Quality of English Language

Author Response

(The authors gave the same response as above.)

Round 2

Reviewer 1 Report

Comments and Suggestions for Authors

Since the authors' responses to the Reviewer's comment were adequately addressed, I accept this manuscript for the publication in Biomolecules.